# Evaluation of a Maternal Plasma RNA Panel Predicting Spontaneous Preterm Birth and Its Expansion to the Prediction of Preeclampsia

**DOI:** 10.3390/diagnostics12061327

**Published:** 2022-05-27

**Authors:** Carl Philip Weiner, Howard Cuckle, Mark Louis Weiss, Irina Alexandra Buhimschi, Yafeng Dong, Helen Zhou, Risa Ramsey, Robert Egerman, Catalin Sorin Buhimschi

**Affiliations:** 1Department of Obstetrics and Gynecology, Kansas University Medical Center, Kansas City, KS 66160, USA; yafengdong0228@163.com (Y.D.); huizhoulp@gmail.com (H.Z.); 2Rosetta Signaling Laboratory, Phoenix, AZ 85018, USA; 3Faculty of Medicine, Tel Aviv University, Ramat Aviv, Tel Aviv 6934206, Israel; h.s.cuckle@leeds.ac.uk; 4Departments of Anatomy and Physiology & Midwest Institute of Comparative Stem Cell Biology, Kansas State University, Manhattan, KS 66506, USA; weiss@vet.k-state.edu; 5Department of Obstetrics and Gynecology, University of Illinois-Chicago, Chicago, IL 60612, USA; irina@uic.edu (I.A.B.); csb01@uic.edu (C.S.B.); 6Office of Clinical Research, University of Tennessee Health Science Center, Memphis, TN 38163, USA; rramsey@uthsc.edu; 7Department of Obstetrics and Gynecology, University of Florida, Gainesville, FL 32611, USA; egerrs@ufl.edu

**Keywords:** preterm birth, extreme prematurity, preterm labor, preeclampsia, early onset preeclampsia, plasma RNA, plasma transcriptome, prognostication, screening

## Abstract

Preterm birth is the principal contributor to neonatal death and morbidity worldwide. We previously described a plasma cell-free RNA panel that between 16 and 20 weeks of pregnancy had potential to predict spontaneous preterm birth (sPTB) ≤ 32 weeks caused by preterm labor (PTL) or preterm premature rupture of membranes (PPROM). The present study had three objectives: **(1)** estimate the RNA panel prognostic accuracy for PTL/PPROM ≤ 32 weeks in a larger series; **(2)** improve accuracy by adding clinical characteristics to the predictive model; and **(3)** examine the association of the RNA panel with preeclampsia. We studied 289 women from Memphis TN prospectively sampled 16.0–20.7 weeks and found: (1) *PSME2* and *Hsa-Let 7g* were differentially expressed in cases of PTL/PPROM ≤ 32 weeks and together provided fair predictive accuracy with AUC of 0.76; (2) combining the two RNAs with clinical characteristics improved good predictive accuracy for PTL/PPROM ≤ 32 weeks (AUC 0.83); (3) *NAMPT* and *APOA1* were differentially expressed in women with ‘early-onset preeclampsia’ (EOP) and together provided good predictive accuracy with AUC of 0.89; and (4) combining the two RNAs with clinical characteristics provided excellent predictive accuracy (AUC 0.96). Our findings suggest an underlying common pathophysiological relationship between PTL/PPROM ≤ 32 weeks and EOP and open inroads for the prognostication of high-risk pregnancies.

## 1. Introduction

Spontaneous preterm birth (sPTB) is the leading cause of mortality in children under 5 years of age. Among survivors, complications of prematurity are a major cause of short- and long-term disability with life-long consequences for affected individuals, their families, and society at-large [1,2,3,4,5]. The proposition here is that predicting the pregnancies likely to result in sPTB could lead to better outcomes even though current therapeutic options to prevent sPTB ≤ 32 weeks are limited. Currently, an obstetrical history of prior PTB and the sonographically measured cervical length (CL) at 18–22 weeks of pregnancy are the main variables used to select at-risk individuals for the available therapeutic options [6].

We reported [7] the differential expression of five plasma cell-free (PCF) RNAs (*PSME2*, *NAMPT*, *APOA1*, *APOA4,* and *Hsa-Let 7g*) at 24 weeks of pregnancy in asymptomatic women who ultimately experienced spontaneous preterm labor and delivery (sPTL) or preterm premature rupture of membranes (PPROM) leading to sPTB ≤ 32 weeks. Importantly, these RNAs were selected because, in silica, they interacted with a subset of myometrial RNAs known to be differentially expressed in women who experienced sPTB ≤ 32 weeks due to sPTL [8]. An independent 40-patient case–control study included in discovery report [7], confirmed differential expression at 24 weeks of four of the five RNAs for sPTL/PPROM ≤ 32 weeks (*APOA1* approached but did not reach significance). We further demonstrated that overexpression of *APOA4* in immortalized human pregnant myometrial cells increased cell contraction frequency, and that this response was blocked either by antisense RNA or by blocking the putative downstream target of *APOA4* [7].

Despite great care to select discovery stage subjects free of comorbidities, gene mapping revealed that hypertension was the number one disease associated with the identified RNAs [7]. This led us to hypothesize that one or more of these RNAs might also predict pregnancy-associated hypertension, e.g., preeclampsia. Thus, the present study had three objectives: **(1)** to retest the prognostic accuracy of the RNA panel at 16–20 weeks for sPTL/PPROM ≤ 32 weeks in a prospectively enrolled cohort of women inclusive of common pregnancy complications; **(2)** to improve predictive modeling for sPTL/PPROM ≤ 32 weeks by combining RNAs and clinical characteristics; and **(3)** to evaluate the association of these RNAs with pregnancy-associated hypertension by testing their relationship to ‘all preeclampsia’ and early-onset preeclampsia (EOP) cases. 

## 2. Materials and Methods

### 2.1. Clinical Methods

Enrollment and pregnancy care took place in Memphis TN, USA, between 2005 and 2019. Multiple specimens were obtained at enrollment (16–20 weeks’ gestation) including maternal plasma [4 mL vacuum tubes coated with 7.2 mg K_2_EDTA]. Samples were obtained biweekly after enrollment until delivery when an umbilical cord blood was obtained. Samples were placed on wet ice and centrifuged within 60 min. The resulting plasma was aliquoted in 500 µL fractions and stored at −80 °C except when the deidentified samples were transferred on dry ice to the University of Kansas Pregnancy Biobank maintained by the Principal Investigator (CPW). All clinical records were extracted contemporaneously and entered into a computerized database. Entries were checked for accuracy on a randomized basis. The records were then finalized within 6 months of the last participant’s delivery and all identifiers removed from the database before distribution to investigators. The identifying linkage and original records are maintained on a secure server at the University of Tennessee Health Science Center in Memphis under the control of an onsite investigator (RR).

Only singleton pregnancies were included. Pregnancy dating used the last menstrual period (LMP) when available confirmed by an ultrasound performed ≤20 weeks. If the two differed by more than 11 days or there was no LMP, the ultrasound-derived gestation was used. Information was collected on maternal characteristics including race, ethnicity, education, smoking status, parity, gravidity, previous PTB and the gestation at which it occurred, maternal age, and maternal weight. The MAP was obtained at the time of sampling. Those women screened by their providers with CL had the measurement performed 18–22 weeks’ gestation, or about 2–6 weeks after blood sampling.

Women who experienced spontaneous labor and delivered either ≤32 weeks or <37 weeks or experienced PPROM and delivered (whether by spontaneous labor or obstetrically indicated induction of labor) ≤32 weeks or <37 weeks constituted the PTB cases. Likewise, women diagnosed with preeclampsia by their managing physicians employing standard criteria and delivered <34 weeks whether spontaneously or for medical or obstetrical indications were classified as EOP. All other women diagnosed with preeclampsia were included in the ‘All Preeclampsia’ group.

### 2.2. Laboratory Methods

Laboratory personnel were blinded to all pregnancy outcomes.

#### 2.2.1. RNA Extraction

PCF RNA was extracted by Rosetta Signaling Laboratory (Phoenix, AZ, USA) using a proprietary method. The mean total RNA extracted was 15.9 ± 2.2 µg/mL (± standard deviation). RNA yield was assessed with a Nano spectrometer (NanoDrop Technologies, Wilmington, DE, USA) and RNA integrity confirmed by the Agilent Bio-analyzer (Agilent, Santa Clara, CA, USA).

#### 2.2.2. qRT-PCR Assays

mRNA RT: The RNA was diluted with a master mix including dNTP mix, Omniscript Reverse Transcriptase and Random Primer (Invitrogen, Carlsbad, CA, USA) and converted into cDNA at 37 °C for 60 min per manufacturer instructions.

miRNA RT: miRNA was polyadenylated (Invitrogen NCode miRNA First-Strand cDNA Synthesis Kit, ThermoFisher, Waltham, MA, USA) and reverse transcribed to generate the first strand of cDNA according to the manufacturer’s protocol.

#### 2.2.3. Preamplification and qPCR

Multiplex qPCR reactions were performed using SYBR Green Supermix (ThermoFisher) and the ViiA 7 Real-Time PCR System (ThermoFisher). The primers were custom designed and synthesized by Integrated DNA Technologies (IDT, Coralville, IA, USA). Information about the primer sequences is available from the authors. Probe sets in each reaction well included primers for the biomarker, normalization, and spike genes so that all were run in the same reaction to minimize assay variation. The 1 µL RT samples were prepared for the preamplification Mix Reaction and underwent 12 cycles. Then, 2 µL of preamplification cDNA samples were diluted into 10µL PCR reaction mix followed by RT PCR. Threshold cycles (CT values) of qPCR reactions were extracted using QuantStudio™ Software V1.3 (Applied Biosystems, Foster City, CA, USA). The delta-delta CT method was used to calculate RNA expression and then normalized as described [7].

### 2.3. Statistics

The expression of each RNA, as well as CL and MAP, were converted into a multiple of the median (MoM) derived from unaffected pregnancies using either the overall median or a regression if there was an association between expression and gestational age or maternal weight. The distribution of log MoMs in cases and non-cases was compared using the Wilcoxon Rank Sum Test and maternal characteristics were compared using a Chi-square test. RNAs and factors with a statistically significant difference (defined as *p* < 0.05 in two tailed testing) were subject to logistic regression analyses alone and in combination—CL was included in PTB analyses and MAP in preeclampsia. On occasion, RNA expression level fell below detection for *NAMPT* and *APOA1*. In those instances, separate regression equations were derived for the detected RNA. Each regression yielded an AUC to obtain a measure of performance across all possible classification thresholds [9].

The regression equation was used to derive a probability, *p*, where *p* = y/(1 − y), y = e^x^ and x is a linear function of log MoMs and maternal characteristics. Among controls, the 90th, 80th, and 70th percentiles of *p* were used to determine the observed DR for three fixed FPRs (10%, 20%, and 30%) since an acceptable FPR could vary if the cost of misclassification was part of model selection [10]. The number of cases per clinical group varied by case definition (PTB < 37 w, sPTB ≤ 32 w, preeclampsia and EOP < 34 w), but the total sample size (case + non-case) remained constant unless undertaking an analysis of a subset (e.g., CL).

## 3. Results

### 3.1. Description of the Cohort

In total, 305 women with singleton pregnancies were prospectively enrolled and sampled biweekly 2005–2009 to develop a pregnancy biobank funded by the US Centers for Disease Control 2005–2010 (STARD Diagram, Appendix A). By design, about half the women enrolled had a history of prior PTB. The final cohort included 289 women (6 never sampled (2.0%), 6 lost to follow-up (2.0%), and 4 clinical exclusions (1.3%): 1 “spontaneous” chorioamnionitis at 19 weeks suspected of self-inducement, 1 PPROM 48 h after elective cerclage performed at 24 weeks, 1 placental abruption at 28 weeks, and 1 appropriately grown IUFD at 35 weeks). An additional 9 women (1 case of sPTB ≤ 32 weeks and 8 term deliveries) received prophylactic progesterone after RNA testing that could have altered clinical outcome [11]; in a subset analysis, they were excluded. No other subject received a cerclage, and no subject was shown to be abusing illicit drugs. Table 1 provides additional demographic information which reflected the population served by the medical center. Most women self-identified as non-Hispanic Black (246, 85%). The overall rate for PTB < 37 weeks was 25% and for PTB ≤ 32 weeks 10%. About one-third of PTBs were preceded by PPROM and about one-fifth by EOP. The 192 pregnancies that did not result in PTB or preeclampsia were used as ‘non-cases’ when comparing results in either adverse outcome.

### 3.2. Prediction of sPTL/PPROM ≤ 32 Weeks

There was a statistically significant relationship between sPTL/PPROM ≤ 32 weeks and non-Hispanic Black race (*p* < 0.05), a prior history of PTB (*p* < 0.0001) (Table 2) and, among those with a history, if it was ≤ 32 weeks’ gestation PTB (*p* < 0.01) (Table 3). There was no significant relationship between sPTL/PPROM ≤ 32 weeks and maternal age or parity in those with no history of PTB. In the subsequent logistic regression analysis, the two significant factors on prior PTB were combined into a single variable—the gestational age of the earliest PTB or 40 weeks if there were none. 

Among the five PCF RNAs tested [7], *PSME2* (*p* < 0.001) and *Hsa-Let 7g* (*p* < 0.0001) were differentially expressed in cases who experienced sPTL/PPROM ≤ 32 weeks (Table 4). *APOA4* (*p* = 0.092), *APOA1* (*p* = 0.43), and *NAMPT* (*p* = 0.12) did not achieve significance in this cohort. While not part of the study design, 154 women were screened by their managing obstetricians by measurement of a CL 2–6 weeks after the enrollment blood sample (35 sPTB cases, 12 of which were sPTL/PPROM ≤ 32 weeks, and 119 term deliveries), and CL was evaluated as a predictive variable. Those cases with sPTL/PPROM ≤ 32 weeks had significantly shorter CL measurements (*p* < 0.0001, Table 4).

Multiple regression used to model sPTL/PPROM ≤ 32 weeks using *PSME2* and *Hsa-Let 7g* provided an AUC of 0.76 [95% CI 0.65–0.87] with corresponding detection rates (DRs) of 50%, 57%, and 70% for fixed false positive rates (FPRs) of 10%, 20%, and 30% (Table 5). Adding maternal characteristics of prior sPTB and self-identified race to the predictive model increased the AUC to 0.83 [95% CI 0.74–0.92] with DRs of 63%, 77%, and 77% for the same fixed FPRs (Table 5). Removing the data subset obtained from the 9 women treated with progesterone from the predictive model had no impact on the predictive power of the model: AUC (0.85, 95% CI 0.75–0.94). The best fit logistic prediction equation for sPTL/PPROM ≤ 32 weeks was: x = 1.4677 + 0.5732*log10 (*PSME2* multiples of the median (MoM) + 0.8268*log10 (*Hsa-Let 7g* MoM) −1.8931 if race was other than non-Hispanic Black −0.1119* earliest gestation of prior PTB or 40 weeks if none.

Mentioned above, CL was obtained in a subset of cases (*n* = 12) with sPTL/PPROM ≤ 32 weeks. Regression analysis modeling yielded an AUC of 0.86 (95% CI 0.75–0.96) (Table 5). Adding CL to *PSME2*, *Hsa-Let 7g* and maternal characteristics to the predictive model generated an AUC of 0.91 [95% CI 0.84–0.98] with DRs of 67%, 92%, and 92% for the same fixed FPRs, (Table 5). There was a nonsignificant negative correlation between CL and *Hsa-Let 7g* (r = −0.54955, *p* = 0.0642). There was no correlation between CL and any of the five panel RNAs in the 119 non-cases.

PPROM often precedes sPTL. Women with three or more PCF RNA marker expression levels ≥ 1.5 MoM included 64% (18/28) of all PPROM cases and 80% (8/10) of the PPROM cases delivered ≤32 weeks (whether sPTL or indicated). Those women with PPROM and three or more PCF RNA markers above ≥1.5 MoM experienced rupture an average of 5 weeks earlier than women whose PCF RNA levels were closer to the MoM (mean ± SD: 28.4 ± 5 weeks vs. 33.7 ± 2 weeks, *p* < 0.05).

### 3.3. Prediction of All sPTL/PPROM < 37 Weeks

When all cases of sPTB< 37 weeks were considered, only *PSME2* was differentially expressed compared to non-cases (*p* < 0.05) (Table 4). A regression model built using *PSME2* generated an AUC for sPTB < 37 weeks of 0.58 [95% CI 0.50–0.66] with DRs of 21%, 38%, and 47% for fixed FPRs of 10%, 20%, and 30% (Table 5). Adding significant maternal characteristics of race, smoking, and prior sPTB to the model improved predictive power: AUC to 0.77 [95% CI 0.70–0.83] with DRs of 33%, 59%, and 73% for the same fixed FPRs (Table 5). The best fit prediction logistic equation for PTB < 37 weeks was: x = 2.9733 + 0.1964*log_10_(*PSME2* MoM) −1.3671 if race other than non-Hispanic Black −0.8335 if a smoker −0.1111* earliest gestation of previous PTB or 40 weeks if none.

CL was significantly reduced in all PTBs cases (*p* < 0.002, Table 4), but its inclusion in the predictive model had minimal impact on predictive power: AUC (0.79, 95% CI 0.70–0.88) (Table 5). In cases, there were significant interactions between CL and RNAs: negative correlations were found between CL and *APOA4* (r = −0.45261, *p* = 0.0063) and *Hsa-Let 7g* (r = −0.41581, *p* = 0.0130). In contrast, there was no interaction between CL and any RNAs in non-cases.

## 4. Prediction of Early-Onset Preeclampsia < 34 Weeks (EOP)

Both *NAMPT* and *APOA1* were differentially expressed in EOP cases leading to preterm birth (indicated or spontaneous) (*p* < 0.02 and *p* < 0.05, respectively) (Table 6). Combining the two RNAs in a predictive model yielded an AUC of 0.89 [95% CI 0.79–0.98] and DRs of 50%, 67%, and 100% for the same fixed FPRs (Table 7). Adding parity (Table 8) to the model produced a small increase in predictive power: AUC to 0.94 [95% CI 0.89–0.99] (Table 7).

Pre-pregnancy hypertension is associated with both preeclampsia and EOP [1]. Adding the mean arterial pressure (MAP) obtained at the time of sampling to the RNA-only model increased predictive power: AUC to 0.92 [95% CI 0.84–0.99], with DRs of 67%, 83%, and 100% for the same fixed FPRs (Table 7). Lastly, the best predictive model combined *NAMPT* and *APOA1* expression with MAP and parity and yielded an AUC to 0.96 (95% CI 0.92–1.0). The model for prediction of EOP weeks was: x = −7.900 + 2.2726*log_10_(*NAMPT* MoM) + 0.8267*log_10_(*APOA1* MoM) + 3.6091 if nulliparous or −7.9772 if parity is 1 or 2.

### Prediction of All Preeclampsia

*NAMPT* (*p* = 0.022) and MAP (*p* = 0.0017) were differentially expressed in cases that developed preeclampsia (*p* < 0.05) at some point during the pregnancy (Table 6). Predictive models including *NAMPT* yielded an AUC of 0.65 [95% CI 0.53–0.76] with DRs of 30%, 35%, and 39% for the same fixed FPRs, while models including MAP yielded an AUC of 0.68 [95% CI 0.57–0.80] with DRs of 32%, 41%, and 55% for the same fixed FPRs (Table 7). Models that combined *NAMPT* with parity increased the AUC to 0.72 (95% CI 0.60–0.84). Models that combined *NAMPT* and MAP increased predictive power: AUC to 0.77 [95% CI 0.68–0.86] with DRs of 38%, 52%, and 62% for the same fixed FPRs (Table 7). The combination of *NAMPT* with both MAP and parity in the predictive model increased the AUC to 0.82 (95% CI 0.72–0.91). The best fit equation for the prediction of preeclampsia was: x = −2.1134 + 1.0161*log_10_(*NAMPT* MoM) + 21.5283*log_10_(MAP MoM) + 0.8873 if nulliparous or −0.8217 if parity is 1 or 2.

## 5. Discussion

To address our first goal, a panel of five PCF RNAs previously found at 24 weeks to be differentially expressed by microarrays in women destined for sPTL/PPROM ≤ 32 weeks was re-tested in samples obtained 16–20 weeks in a prospective cohort, and two of five RNAs were confirmed to be differentially expressed [7]. Our second goal was to improve predictive modeling by including clinical variables such as prior PTB, parity, etc. In general, we found that adding specific clinical variables improved the predictive accuracy. Since gene mapping of the RNAs revealed hypertension as the number one associated condition [7], our third goal was to expand our analysis to determine whether these RNAs would predict preeclampsia. The current study of samples not part of discovery confirmed that models containing four of the five previously identified RNAs can predict the three most common pregnancy complications leading to PTB ≤ 32 weeks with good to excellent accuracy. The predictive models described here included both RNA and patient biographic information and are better predictors of sPTB than the current standard of care which uses predominantly clinical history and CL [6]. Specifically, the AUC for a predictive model using *PSME2, Hsa Let-7g* plus maternal characteristics for sPTL/PPROM ≤ 32 weeks was 0.83, providing confirmation of RNAs previously discovered to predict sPTB [7]. To resolve questions raised by previous gene mapping of these RNAs to hypertension [7], we evaluated one or more of the five RNAs that could be used to predict preeclampsia. Critically, we found that predictive models using *NAMPT* and *APOA1* plus MAP and maternal characteristics achieved excellent predictive power: AUC of 0.96 for EOP. This confirmed our speculation that the RNAs originally found to be predictive of sPTL ≤ 32 weeks were actually effective predictors of EOP which was not part of discovery. To summarize the key findings: models using RNAs from blood sampled at 16–20 weeks of pregnancy together with maternal characteristics and medical history had good to excellent power to predict PTB ≤ 32 weeks whether sPTL or EOP.

The AUC provides an aggregate measure of performance across all possible classification thresholds [9]. A test with an AUC between 0.90 and 1.00 is considered excellent, one between 0.8 and 0.9 good, 0.7–0.8 fair, 0.6–0.7 poor, and 0.5–0.6 not useful [12]. In addition to AUC, we estimated performance with the detection rates (DR) for three fixed false positive rates (FPRs) (10%, 20%, and 30%) since an acceptable FPR could vary if the cost of misclassification was part of model selection [10]. The comparator group in this analysis had neither PTB nor preeclampsia. In clinical practice, a large proportion of preeclampsia cases deliver at term and these study cases could have been included as controls. However, this design was chosen to avoid any potential confounding in markers for the two adverse outcomes.

A strength of the current study is the cohort design consisting of pregnancies managed at a single, tertiary center and associated with a low dropout rate (5%). The design enabled us to have reasonably large number of sPTL/PPROM ≤ 32 weeks cases. Another strength is the methodology for plasma RNA extraction that increases the total RNA yield per milliliter of plasma to microgram quantities rather than the nanogram amounts achieved by existing commercial kits [13,14].

There were also study limitations. For example, the cohort was intentionally enriched during enrollment with women at high risk for sPTB based on their obstetric history. Additionally, while the racial demographics of the enrolled cohort mirrored the population served by the health care center in which it was based, 85% of participants self-identified as non-Hispanic Black. These two factors raise questions about the ability of these findings to be applied to other populations, especially if the conventional wisdom that PTB ≤ 32 weeks is a syndrome with multiple etiologies is correct. However, since this study follows on from work that utilized an equal racial mix and confirmed the five RNAs discovered and four of five RNAs in a case–control study [7], the findings are not easily discounted. Furthermore, the predictive accuracy for sPTL/PPROM ≤ 32 weeks appeared unaffected by parity. Finally, the finding that RNAs collected at 16–20 weeks of pregnancy may identify those destined to suffer sPTB ≤ 32 weeks would seem to argue against the accuracy of such conventional wisdom of there being numerous causes.

A weakness of the current study is the small number of EOP cases requiring delivery before 34 weeks. Another weakness was that a single cohort was studied limiting geographic and patient diversity. The final study limitation was the modest number of historical variables collected. These samples were collected in 2005–2009, and in the intervening years the paper medical record was replaced by a computerized medical record, and the old records stored off location and poorly accessible for post hoc abstraction. While the study did include major risk factors such as prior PTB, study personnel may not have consistently recorded other now recognized risk factors for PTB such as chronic hypertension, or untreated type 2 diabetes or gestational diabetes. It will be important moving forward to re-evaluate these results in future broader, more diverse patient cohorts because the study cohort here was by-design a high-risk group, and because of that we selected AUC as the main performance indicator, rather than negative/positive predictive values which are impacted by disease prevalence.

In current standard of care, PTB predictors used include historical, biological, and ultrasound-based markers [15,16]. Clinical history is a poor-to-fair predictor of PTB by AUC [17], and the measurement of cervical fetal fibronectin (fFN) or CL at 16–22 weeks yields AUCs for sPTL/PPROM ≤ 32 weeks of 0.51–0.54 and 0.51–0.58, respectively [6]. In other large cohorts [18] AUCs for CL measurement at 18–22 weeks range from 0.51 to 0.76 for sPTB < 37 w. It might be argued the popularity of these tools reflects the lack of options.

Transvaginal ultrasound CL was not part of the study design, and the women selected for CL measurement by their caregivers may represent a group perceived as at highest risk. While the predictive models for sPTL/PPROM ≤ 32 weeks achieved in the current study were similar when CL or PCF RNAs were considered individually, their combination in the model appeared to enhance predictive power, raising the AUC for sPTL/PPROM ≤ 32 weeks from 0.83 to 0.91. While there was no correlation between CL and any of the five RNAs in the non-cases, CL was inversely correlated with *Hsa-Let-7g* expression in all PTB cases. This interesting finding may reflect how the five RNAs were themselves were identified—they were differentially expressed RNAs discovered on microarray with the in silica potential to interact with myometrial preterm birth initiator RNAs previously reported [7,8]. It is possible that *Hsa-Let-7g,* which was predictive of sPTL/PPROM ≤ 32 weeks, may also be predictive of premature cervical ripening. Future cohorts will allow us to explore the possible relationship between CL and *Hsa-Let-7g,* and to determine whether CL measurement at 18–22 weeks truly enhances the prognostic accuracy for sPTB achieved by the combination of *PSME2*, *Hsa-Let-7g*, maternal characteristics, and clinical markers. Even if the RNA markers alone generate an AUC similar to CL for sPTL/PPROM ≤ 32 weeks, the information would be available at least one month earlier than CL, and its combination with *NAMPT* and *APOA1* has the potential to provide from a single maternal blood sample prognostic information on the three major pregnancy disorders leading to sPTB (sPTL, PPROM, and EOP < 34 weeks) rather than just two like CL.

Progesterone supplementation was neither part of our study design nor widely used in Memphis during the enrollment period. Only 9 of the 305 enrolled women received weekly intramuscular injections of 17-OH progesterone. One patient suffered sPTL/PPROM ≤ 32 weeks, and 8 delivered at term. Their exclusion did not significantly alter the resulting AUC. Future cohorts will enable a direct examination of what effect, if any, progesterone supplementation has on the predictive models described here. Other potential prophylactic therapies for PTB, such as DHA, were not used in Memphis TN at the time the cohort was assembled.

Increased expression of *PSME2* and *Hsa-Let-7g* were predictors of sPTL/PPROM ≤ 32 weeks in both our initial [7] and current studies, and in silica analyses suggest that each of the five RNAs has the potential to alter myometrial cell contraction frequency [7]. However, their expression in either plasma or myometrial cells is unlikely the proximate cause of sPTL/PPROM ≤ 32 weeks since several women delivered at term despite high expression levels of all five panel markers (data not shown). Interestingly, the predictive model performs similarly in subjects who self-identified as either White or non-Hispanic Black. It may be these RNAs that interfere with myometrial quiescence and create an environment conducive to sPTB but does not actually ‘light the fuse’.

Other work has found biomarkers that predict sPTB. For example, insulin-like growth factor binding protein 4/sex hormone binding globulin (IBP4/SHBG) measured by mass spectroscopy at 19–20 weeks is currently commercialized as a sPTB screening test [15,19,20]. Recently, a study made up of 847 patients [21] reported an AUC of 0.76 when the ratio was combined with numerous maternal characteristics (the highest AUC for this panel to date).

Ngo et al. [22] reported a case–control validation study of a seven PCF RNA panel on 5 Black women who experienced PTB (mean 30.6 ± 2.4 w) and 18 Black women with at term delivery (mean 38.7 ± 0.5 w). In total, 17 of the 23 women were sampled at or after 24 weeks. Their predictive mode of RNAs plus clinical characteristics yielded an AUC of 0.81. Ngo’s study design may inadvertently have decreased the likelihood of achieving an early pregnancy screening test since all but 1 of their discovery subjects was sampled at or after 24 weeks. This later sampling period may explain why none of the RNAs in their panel overlapped with those used here [7].

Cook et al. [23] conducted both discovery (*n* = 13 PTB/40 controls) and validation (*n* = 14 PTB/124 controls) studies of maternal plasma microRNA (miR) expression across gestation. They validated 9 miRs differentially expressed at 18–21.9 weeks in women destined for sPTB < 34 weeks. Three of the 9 individual miRs provided models with very good predictive power with AUCs ranging from 0.80 to 0.87. One of their markers was Let-7a that might behave similarly to Let-7g in the current study, and none of the other 8 miRs overlapped with 10 differentially expressed miRs we previously reported [7]. Jehan et al. [24] used a multiomics approach to build predictive models including maternal characteristics (81 women cohort, 39 with PTB < 37 weeks and 42 controls sampled at 8–20 weeks (median 13.6 weeks)). Using machine learning to build their predictive models, their best model integrating the three ‘omics’ yielded an AUC of 0.83. The individual ‘omics’ AUC values were 0.73 (transcriptomics), 0.59 (metabolomics), and 0.75 (proteomics). None of their listed markers overlapped the current RNAs.

As it relates to EOP, the current study was based upon our previous pathway analysis that associated these RNAs with hypertension [7] and the results are encouraging. There have been other notable efforts to predict EOP. For example, a first trimester risk model was described by O’Gorman et al. [25,26] and subsequently confirmed and refined in large multicentered trials yielding AUCs between 0.90 and 0.92 [27]. This is the benchmark test for EOP. To achieve maximum predictive accuracy, the test requires maternal obstetrical and medical histories, the measurement of two serum proteins, a standardized blood pressure, and a first trimester maternal uterine artery Doppler resistance measurement. In some health systems, the ultrasound and patient visit are considered additional to routine prenatal care. In contrast, in the current study, a blood sample obtained at 16–20 weeks together with MAP and parity generated a predictive model achieved an AUC of 0.96 for EOP. If the present study is subsequently confirmed and deployed, EOP and sPTB predictive testing could be integrated into routine prenatal care without requiring additional visits.

Several investigators have explored using the plasma transcriptome to predict preeclampsia. Farina et al. [28] were the first to explore the use of RNA from a blood sample by studying at 10–14 weeks from 11 women who ultimately experienced preeclampsia with 88 control subjects. Higher multiples of the median values than expected were found for endoglin, fms-related tyrosine kinase 1, and transforming growth factor-β1. Lower multiples of the median values were found for placental growth factor and placental protein 13. Endoglin fms-related tyrosine kinase 1 and transforming growth factor-β1 had the best discriminant power. Del Vecchio et al. [29] described a 1st trimester discovery study using mRNA RNAseq on plasma samples from 9 normal pregnancies, 5 pregnancies that developed preeclampsia, and 3 that developed gestational hypertension. They reported 42 differentially expressed PCF RNAs in the mid 2nd trimester. A regression model using five RNAs selected for ‘pregnancy hypertension’ (preeclampsia and gestational hypertension) yielded good predictive power with an AUC of 0.86, and like the present study, their proposed RNA panel included *NAMPT* (the remaining markers were *MMP8*, *SRPK1*, *S100A9*, and *S100A8*). In the current study, *NAMPT* plus MAP and parity generated a model with similar predictive power: AUC of 0.82 for all preeclampsia cases. Nampt protein levels have been reported to be either elevated or reduced in women with preeclampsia [30]. Tarca et al. [31] reported an mRNA discovery study using whole blood and microarrays. Their best RNA panels provided fair predictive power with AUCs in the mid 0.70s. None of their proposed RNAs overlapped with those reported here. Munchel et al. [32] reported a discovery study of blood samples obtained from symptomatic EOP cases plus age-matched non-cases followed by a 10-women validation sample that employed machine learning to generate the predictive model. Thirty RNAs were deemed of interest. The best AUC was 0.96. The application of machine learning to a small sample is challenging, and their results may reflect overfitting. None of their proposed RNAs overlap those reported here.

Rasmussen et al. reported an excellent study of the maternal plasma transcriptome combining samples across gestation from multiple existing and diverse pregnancy biobanks [33]. Relevant to the present study, Rasmussen et al. included a case–control study with 72 cases of preeclampsia and 452 non-cases selected from 2 independent cohorts that included 31 non-cases with chronic hypertension and 19 cases with gestational hypertension along with an unspecified number of preterm births. Differentially expressed RNAs identified by RNAseq were subject to two-sided Spearman correlation testing to identify signatures that best separated cases and non-cases. Seven RNA were identified: *CLDN7*, *PAPPA2*, *SNORD14A*, *PLEKHH1*, *MAGEA10*, *TLE6,* and *FABP1*. Logistic regression to estimate the probability of preeclampsia yielded an AUC of 0.82. Interestingly, the inclusion of clinical variables did not enhance the predictive power of their model, in contrast to what we report here.

Most recently, Moufarrej et al. [34] identified 544 differentially expressed PCF RNAs altered across gestation and postpartum. Focusing on early pregnancy, they identified 18 RNAs differentially expressed in women who developed preeclampsia that based on regression analysis achieved an AUC of 0.99 (CI: 0.99–0.99). Based upon these preliminary results, they built a model using 18 RNAs and tested it on two other cohorts. The follow-on work achieved fair predictive power with an AUC of 0.71 (CI: 0.70–0.72). The AUC improved slightly to 0.74 (CI: 0.73–0.75) when chronic hypertension and gestational diabetes were added to the model. None of the RNAs they identified overlapped with the current study.

The present study along with each of the RNA studies cited were all built upon the identification of differentially expressed RNAs using the relevant standards of the day. While criteria were employed to identify differentially expressed RNAs for prognostication, all the studies share the virtually complete lack of overlap in predictive RNAs selected. Whether this lack of cross confirmation is of technical or biological origin remains to be determined.

Regardless, all studies require additional testing using cohorts enriched with 1^st^ trimester subjects since the efficacy of low-dose aspirin to prevent EOP is gestation-dependent. The successful identification of a predictive test for both sPTB and EOP would provide caregivers a single test that could identify asymptomatic women likely to develop one or more of the three most important pregnancy disorders, two which already have therapeutic options that may improve outcome [35,36,37]. Additionally, if any of the panels perform similarly to the current studies, they will be highly cost effective [38].

The RNAs used here were selected for sPTL/PPROM ≤ 32 weeks, and four of five RNAs previously identified as predictors of sPTB confirmed [7]. The current study expands upon that work and finds that some of those RNAs were apparently better predictors of EOP, a syndrome phenotypically unlike sPTB ≤ 32 weeks. This begs the question: why? The explanation may lie in the fact that EOP and sPTB share abnormal smooth muscle responsiveness. Plasma RNAs circulate freely, or bound to high density lipoproteins, carrier proteins, or within other transporters such as extracellular vesicles [39]. It is thought that these transporters may function to delay RNA degradation and may provide a mechanism for cell targeting [40]. We hypothesize that sPTB occurs when certain RNAs that are trafficked within transporters and enter myometrial smooth muscle, while EOP occurs when the certain RNAs are trafficked within transporters and enter vascular smooth muscle. We also posit that either RNAs or its transporter may change with ongoing placental development. In our initial case–control study which did not include women with preeclampsia or chronic hypertension [7], *APOA4* was significantly increased in women destined for sPTB ≤ 32 weeks and *APOA1* approached but did not reach significance. In the current cohort study, *APOA4* approached but did not achieve significance for sPTL/PPROM ≤ 32 weeks, but *APOA1* was a strong predictor of EOP suggesting the possibility of disease commonality. In support of this hypothesis, Yoffe et al. [41] reported a 1^st^ trimester RNAseq discovery study of small noncoding RNAs predictive of EOP. Among those differentially expressed were two of the same miRs (*Hsa-Let-7g* and *miR99b*) we identified in 2^nd^ trimester women [7]. Importantly, in our study those RNAs were predictive of sPTL/PPROM ≤ 32 weeks and not EOP. Such puzzles should resolve as experience and knowledge base grows, and when we will be able to: (1) use gestational week-by-week control samples rather than a range of weeks; (2) expand and refine the clinical history variables collected (e.g., prior preeclampsia, chronic hypertension, pre-pregnancy diabetes, etc.); and (3) apply other predictive modeling tools (e.g., machine learning, cost-sensitive learning algorithms) before fixing the predictive model.

## 6. Conclusions

This study and several other studies performed at different gestational ages indicate that PCF RNAs have potential to predict women likely to develop one or more pregnancy disorders leading to sPTB ≤ 32 weeks. The identification of these differentially expressed RNAs in samples collected at 16–20 weeks suggests the likelihood of a woman experiencing sPTB, whether sPTL ≤ 32 weeks, PPROM ≤ 32 weeks and EOP is ‘set’ early in the pregnancy, and that there might be similar pathophysiology between these disorders despite their different clinical presentation. This information might be leveraged to develop more accurate predictive tests for the most common pregnancy complications, and perhaps, the affected pathways might be targeted for novel drug therapy.

## Figures and Tables

**Table 1 diagnostics-12-01327-t001:** Demographic and pregnancy outcomes of the cohort (n = 289).

Variables	Mean ± SD [Range] or n (%)
Maternal Age (years)	24.9 ± 5.1 [16–43]
Gestational Age at Sampling (weeks)	18.3 ± 1.4 [16.0–20.9]
Race and Ethnicity	
Non-Hispanic White	32 (11)
Non-Hispanic Black	245 (85)
Hispanic	12 (4)
Parity	
Nulliparous	49 (17.0)
Grand multiparous	20 (6.9)
Elective abortions	
At least one	44 (15.3)
Two or more	18 (6.2)
Prior PTB	151 (63% of multipara)
Tobacco use	79 (27.3)
Pregnancy Outcomes	
sPTB < 37 weeks	73 (25.3)
sPTB ≤ 32 weeks	30 (10.4)
Preeclampsia (all)	22 (7.6)
Early-onset preeclampsia < 34 weeks	6 (2.1)

Abbreviations: SD, standard deviation; PTB, preterm birth.

**Table 2 diagnostics-12-01327-t002:** Relationship between maternal characteristics and preterm birth in current pregnancy.

Characteristic	Outcome, % (n)			Significance	
	*PTB < 37 w* *(n = 73)*	*sPTB* *≤ 32 w* *(n = 30)*	*Non-Cases* *(n = 192)*	*PTB < 37 w* vs. *Non-Cases*	*sPTB**≤ 32 w* vs. *Non-Cases*
Race				<0.01	<0.05
Non-Hispanic Black	94% (69)	97% (29)	81% (155)		
Other	5.5% (4)	3.3% (1)	19% (37)		
Smoking				<0.020	0.100
Yes	16% (12)	17% (5)	31% (60)		
No	84% (61)	83% (25)	69% (183)		
Prior PTB				<0.0001	<0.0001
Yes	79% (58)	83% (25)	43% (83)		
No	21% (15)	17% (5)	57% (109)		
GA at Prior PTB				<0.005	<0.010
≤32 weeks	74% (50)	92% (23)	65% (54)		
33–36 weeks	26% (8)	8.0% (2)	35% (29)		
Parity				<0.020	0.150
Parous	35% (6)	40% (2)	71% (77)		
Nulliparous	65% (9)	60% (3)	29% (32)		

Abbreviations: PTB, preterm birth; GA, gestational age; w, weeks.

**Table 3 diagnostics-12-01327-t003:** Relationship of prior preterm birth with preterm birth in current pregnancy.

# Prior Preterm Births	n	Delivery < 37 weeks % (n)	Delivery ≦ 32 weeks % (n)
0	137	13.9% (19)	3.6% (10)
1	85	38.9% (33)	11.8% (10)
2	35	45.7% (16)	20.0% (7)
≥3	30	56.7% (17)	36.7% (11)

**Table 4 diagnostics-12-01327-t004:** Preterm birth and MoMs of RNA markers, cervical length (CL).

Markers	Median (MoM *)			Significance ^†^	
	All PTB (n)	sPTB ≦ 32 w (n)	Non-Cases (n)	All PTB	sPTB ≦ 32 w
PSME2	1.86 (73)	4.09 (30)	1.00 (192)	0.049	0.0007
NAMPT	1.47 (71)	1.46 (28)	1.00 (182)	0.140	0.120
APOA1	1.66 (72)	1.91 (30)	1.00 (187)	0.480	0.428
APOA4	0.88 (73)	2.01 (30)	1.00 (187)	0.440	0.092
Hsa-Let 7g	2.15 (73)	8.84 (30)	1.03 (192)	0.10	<0.0001
Cervical length	0.89 (35)	0.74 (12)	1.00 (119)	0.0014 ^‡^	<0.0001 ^‡^

* Medians: PSME2 = 0.85; NAMPT = 2.34; APOA1 = 0.16; APOA44 = 2.44; Hsa Let 7g = 10^−0.515+0.0106^^∗days−0.232*kg^; CL = 3.14 + 0.00287*kg. ^†^ Wilcoxon Rank Sum test. ^‡^ 1-tailed test. Abbreviations: kg, maternal weight in kilograms.

**Table 5 diagnostics-12-01327-t005:** (a). Prognostic accuracy for preterm birth (PTB) ≤ 32 weeks and PTB < 37 weeks. (b). Prognostic accuracy for preterm birth (PTB) ≤ 32 weeks and PTB < 37 weeks—CL subgroup.

**(a)**					
**Outcome**	**Model Variables**	**AUC [95% CI]**	**DR (%) for** **Fixed FPR**		
			**10%**	**20%**	**30%**
PTB ≤ 32 w ^†^	Prior PTB, race	0.78 [0.69–0.87]	36	73	80
	PSME2, Let 7g	0.76 [0.65–0.87]	50	57	77
	PSME2, Let 7g, prior PTB, race	0.83 [0.74–0.92]	63	77	77
PTB < 37 w ^†^	Prior PTB, race	0.75 [0.69–0.82]	37	62	73
	PSME2, Let 7g	0.58 [0.50–0.66]	21	38	47
	PSME2, Let 7g, prior PTB, race	0.77 [0.70–0.83]	33	59	73
**(b)**					
**Outcome**	**Model Variables**	**AUC [95% CI]**	**DR (%) for** **Fixed FPR**		
			**10%**	**20%**	**30%**
PTB ≤ 32 w ^†^	CL only ^‡^	0.86 [0.75–0.96]	67	67	67
	PSME2, Let 7g, CL ^‡^, prior PTB, race	0.91 [0.84–0.98]	67	92	92
PTB < 37 w ^†^	CL only ^‡^	0.67 [0.55–0.78]	31	49	57
	PSME2, Let 7g, CL ^‡^, prior PTB, race	0.79 [0.70–0.88]	49	60	77

Derived from logistic regression equations; when one or more items of information is missing, the equation is from the remaining values. Race variable refers to non-White and non-Hispanic ethnicity. ^†^ Sample size: PTB < 37 w (73); PTB ≤ 32 w (30); controls (192). ^‡^ Sample size with CL: PTB < 37 w (n = 35); PTB ≤ 32 w (n = 12); controls (n = 119). Abbreviations: CL, cervix length; AUC, area under the curve; DR, discovery rate; FPR, false positive rate; w, weeks.

**Table 6 diagnostics-12-01327-t006:** Preeclampsia and MoMs of RNA markers, mean arterial blood pressure. (MAP).

Markers	Median (MoM)			Significance ^†^	
	All Preeclampsia (n)	EOP < 34 w (n)	Non-Cases (n)	All Preeclampsia	EOP < 34 w
PSME2	2.31 (24)	3.61 (6)	1.00 (192)	0.290	0.080
NAMPT	1.69 (23)	3.64 (6)	1.00 (182)	0.022	0.013
APOA1	1.62 (24)	15.84 (6)	1.00 (187)	0.410	0.024
APOA4	1.26 (24)	2.19 (6)	1.00 (187)	0.800	0.360
Hsa-Let 7g	0.58 (24)	5.36 (6)	1.03 (192)	0.290	0.160
MAP	1.06 (22)	1.08 (6)	1.01 (173)	0.0017 ^‡^	0.029 ^‡^

Medians: PSME2 = 0.85; NAMPT = 2.34; APOA1 = 0.16; APOA4 = 2.44; Has-Let 7g = 10^−0.515+0.0106*days−0.232^^*kg^; MAP = 75.4 + 0.0409*kg. ^†^ Wilcoxon Rank Sum test. ^‡^ 1-tailed test. Abbreviations: kg, maternal weight in kilograms.

**Table 7 diagnostics-12-01327-t007:** Prognostic accuracy * for preeclampsia (PE) and early onset (<34 w) (EOP).

Outcome	Model Variables	AUC [95% CI]	DR (%) for Fixed FPR		
			10%	20%	30%
EOP < 34 w ^†^	parity	0.83 [0.75–0.91]	17	100	100
	MAP ^‡^	0.73 [0.52–0.94]	33	50	67
	NAMPT, APOA1	0.89 [0.79–0.99]	50	67	100
	NAMPT, APOA1, MAP ^‡^	0.92 [0.84–0.99]	67	83	100
	NAMPT, APOA1, parity	0.94 [0.89–0.99]	17	83	100
	NAMPT, APOA1, parity, MAP ^‡^	0.96 [0.92–1.00]	67	100	100
All preeclampsia ^†^	parity	0.65 [0.54–0.76]	0	33	33
	MAP ^‡^	0.68 [0.57–0.80]	32	41	55
	NAMPT	0.65 [0.53–0.76]	30	35	39
	NAMPT, MAP ^‡^	0.77 [0.68–0.87]	38	52	62
	NAMPT, parity	0.72 [0.60–0.84]	35	48	61
	NAMPT, parity, MAP ^‡^	0.82 [0.72–0.91]	48	71	71

* Derived from logistic regression equations; when one or more items of information is missing, the equation is from the remaining values. ^†^ Sample size: Preeclampsia (24); EOP < 34 w (6); non-cases (192). ^‡^ Available: Preeclampsia (22); EOP < 34 w (6); non-cases (173). Abbreviations: EOP, early onset preeclampsia; MAP, mean arterial pressure; AUC, area under the curve; DR, discovery rate, FPR, false positive rate; w, weeks’ gestation.

**Table 8 diagnostics-12-01327-t008:** Relationships of maternal characteristics (parity) and preeclampsia during current pregnancy.

Characteristic	Outcome, % (n)			Significance	
	*All Preeclampsia* *(n = 24)*	*EOP < 34 w* *(n = 6)*	*Non-Cases* *(n = 192)*	*All Preeclampsia* vs. *Controls*	*EOP < 34 w* vs. *Controls*
Parity				<0.05	<0.01
0	33% (8)	17% (1)	17% (32)		
1–2	29% (7)	0% (0)	57% (109)		
3+	38% (9)	83% (5)	26% (51)		

Early-onset preeclampsia: EOP; weeks: w.

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
