# Peer review of "Evaluation of a Maternal Plasma RNA Panel Predicting Spontaneous Preterm Birth and Its Expansion to the Prediction of Preeclampsia"

_diagnostics, 2022, doi:10.3390/diagnostics12061327_

Round 1

Reviewer 1 Report

Very interesting and promising topic. 

What do you understand by mechanical in the sentence "In current standard of care, PTB predictors used include historical, biological, and mechanical markers" ?

Please explain how your work can be implemented in sPB and PE algorithm mainly in terms of cost-efficiency.

Reviewer 2 Report

In this study authors evaluated a panel of 5 plasma cell free (PCF) RNAs (PSME2, NAMPT, APOA1, APOA4 and Hsa-Let 7g) as prognostic marker for PTL/PPROM and Preeclampsia. They found that PSME2 and Hsa-Let-7g were differentially expressed in cases of PTL/PPROM providing an good predictive accuracy with AUC of 0.76. Moreover, combining PSME2 and Hsa-Let-7g RNAs with clinical characteristics of patients improved PTL/PPROM  predictivity leading to an AUC of 0.83. In addition, NAMPT and APOA1 values coul predict early onset preeclampsia showing an AUC of 0.89. Combining the NAMPT and APOA1 RNAs with clinical characteristics increased AUC to 0.96. 

Manuscript is very interesting and generally well written. Only some points need to be improved. In particular:  

  • Introduction must be implemented since it does not give an idea of the charachteristc of PTB and PE. The altered values of (PCF) RNAs found in PE can be due to the trophoblast immaturity (PMID: 32529396) and vascular dysfunctions (PMID: 34831277) characterising this pathology. Moreover, endometrial inflammation found in preterm birth (PMID: 28466813) could also partecipate to (PCF) RNAs alteration. In fact, increased IFN-ϒ found in preterm delivery (PMID: 12854735) can modulate PSME2 (PMID: 26402139).Moreover, authors should highlight the benefits of using RNA markers in comparison to proteic markers of preterm labor already studied (PMID:32102578, 17992705, 19911417). This would add further importance to the study. 
  • To my opinion, authors should replace "non cases" with controls
